# Critical evaluation of drug response prediction models with DrEval

Judith Bernett [1,6], Pascal Iversen [2,3,6], Mario Picciani [4], Mathias Wilhelm [4,5], Katharina Baum [2,3,7] & Markus List [1,5,7] ✉

Large-scale drug sensitivity screens have enabled training drug response prediction models based on cancer cell line omics profiles to advance personalized medicine. While model performances reported in the literature appear promising, successful translation to the clinic remains limited. In this work, we discuss key obstacles that lead to overly optimistic performance estimates of state-of-the-art models, making it challenging to track progress in the field. To address them, we present DrEval, a pipeline for unbiased, biologically meaningful evaluation of cancer drug response models. DrEval is designed as a living open-source benchmark that integrates baseline and literature models with standardized hyperparameter tuning, statistically rigorous evaluation, cross-study benchmarks, and supports ablation studies and publication-ready visualizations. Using DrEval, we show that deep learning models barely outperform a naive model that predicts only the mean drug and cell line effects, while no complex model outperforms properly tuned tree-based ensemble baselines in relevant settings.

Cancer is a highly heterogeneous class of diseases caused by uncontrolled cell growth and spread. Treatment choice is usually based on broad subtype classification, e.g., by the tissue of origin and a few genetic markers[1]. This approach fails to account for the complexity of cancer, leading to unpredictable and inconsistent treatment outcomes[2]. Multi-omics characterizations of tumor cell lines and corresponding large-scale drug response screens have opened up the possibility of linking treatment resistance to the molecular profile of tumor cells. Examples of drug response screens include the Cancer Cell Line Encyclopedia (CCLE)[3], the Cancer Therapeutics Response Portal (CTRP)[4], and Genomics of Drug Sensitivity in Cancer (GDSC)[5], which exposed cultivated tumor cell lines to cancer drugs, measuring cell survival given various concentrations (viability assays). For drug response prediction, metrics summarizing the fitted, typically sigmoidal dose-response relationship curve are used, such as the half-maximal inhibitory/effective concentration (IC50/EC50) or the area under the curve (AUC).

Predicting drug response using the unperturbed omics profile of these cultivated cell lines has become an active area of research with strategies ranging from statistical models and network analyses to complex deep learning models[6–8]. The approaches can be categorized into single-drug models and global approaches. Single-drug models are trained to predict the drug response for a specific drug and thereby only rely on cell line-characterizing features. In contrast, global models are trained across multiple drugs and utilize drug features, such as their chemical properties, theoretically enabling them to predict response values for unseen drugs. While model performance in the literature appears promising, their incorporation into clinical applications to effectively inform medical decisions has so far been very limited[9].

This gap highlights the need to study and address the challenges that hinder transferable progress in the field[10]. We identify six primary obstacles to meaningful progress in the drug response modeling field (Fig. 1):

[1]Data Science in Systems Biology, Technical University of Munich, Freising, Germany. [2]Department of Mathematics and Computer Science, Freie Universität Berlin, Berlin, Germany. [3]Digital Engineering Faculty, Hasso-Plattner-Institute, University of Potsdam, Potsdam, Germany. [4]Computational Mass Spectrometry, Technical University of Munich, Freising, Germany. [5]Munich Data Science Institute, Garching bei München, München, Germany. [6]These authors contributed equally: Judith Bernett, Pascal Iversen. [7]These authors jointly supervised this work: Katharina Baum, Markus List. ✉e-mail: markus.list@tum.de

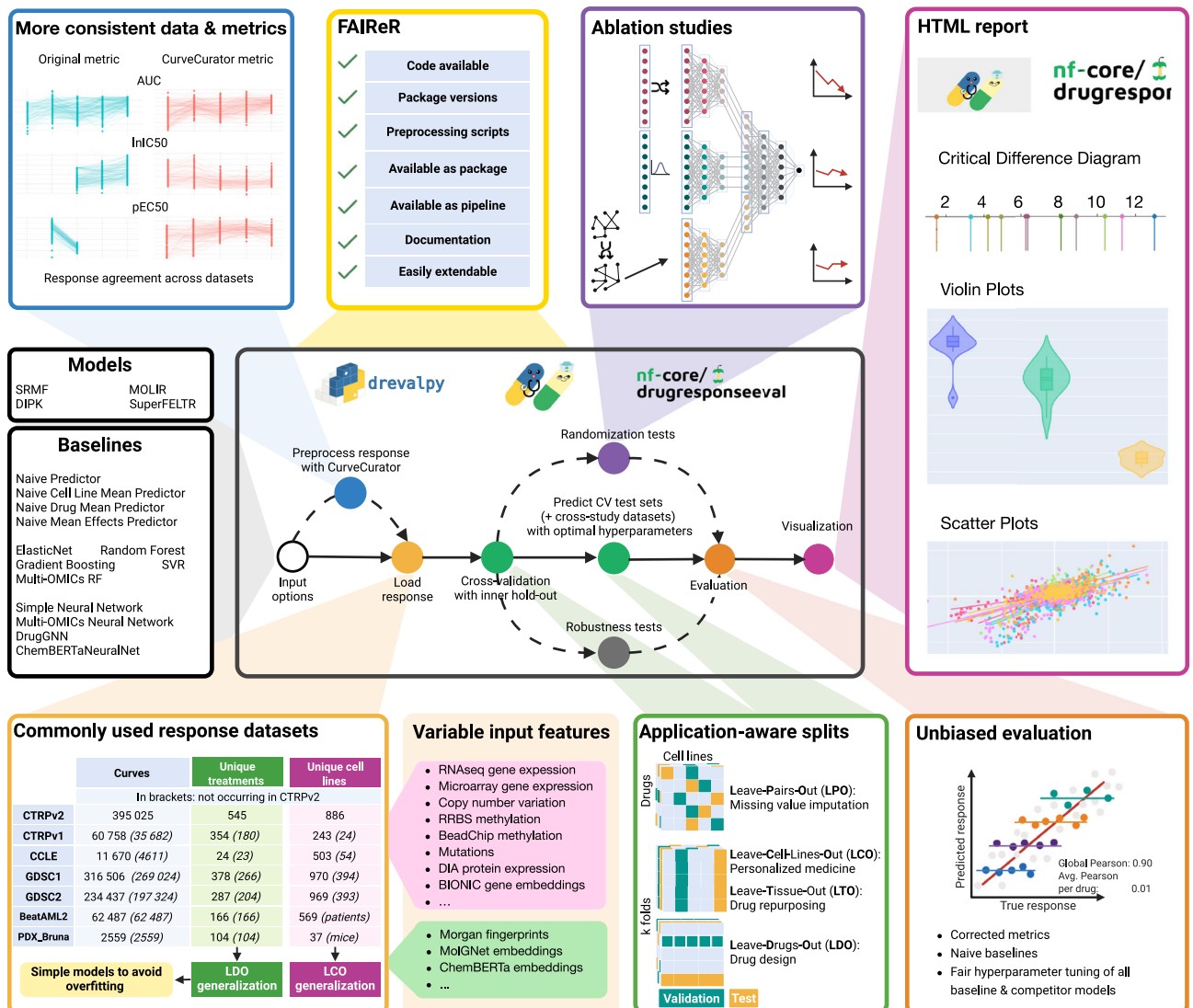

**Fig. 1 | Overview of the DrEval framework.** Via input options, implemented state-of-the-art models can be compared against baselines of varying complexity. We address obstacles to progress in the field at each point in our pipeline: Our framework is available on PyPI and nf-core, and we follow FAIReR standards for optimal reproducibility. DrEval is easily extendable (full tutorial: https://drevalpy.readthedocs.io/en/latest/runyourmodel.html). Custom viability data can be preprocessed with CurveCurator, leading to more consistent data and metrics (full Figure in Supplementary Fig. 4). DrEval supports five widely used cell lines (CTRPv1/2, CCLE, GDSC1/2) and two ex vivo (BeatAML2, PDX_Bruna) drug sensitivity screens with application-aware train/test splits that enable detecting weak generalization. Models are free to use provided or custom cell line- and drug features. The pipeline supports randomization-based feature ablation studies and performs robust hyperparameter tuning for all models. Evaluation is conducted using meaningful, bias-resistant metrics to avoid inflated results from artifacts such as Simpson's paradox. All results are compiled into an interactive HTML report (example: https://dilis-lab.github.io/drevalpy-report/). Created in BioRender. Bernett, J. (2026) (https://BioRender.com/14e1ily), licensed under CC BY 4.0.

1. Reproducibility crisis in drug response prediction: Kapoor and Narayanan[11] postulate that machine learning (ML)-based science is in a reproducibility crisis. They adopt a broader definition of reproducibility, requiring not only computational reproducibility (using the available code and data), but also correct analysis of the data (general reproducibility). An analysis by ref. 12 highlights significant barriers to the reusability of drug response models, citing a lack of modular source code, comprehensive preprocessing information, and adequate documentation. Consequently, drug response prediction is highly affected by the reproducibility crisis, and clear standards of reproducible model sharing are needed.

2. Data leakage and weak generalization: Data leakage occurs when information from the test set (inadvertently) influences the training process, leading to optimistically biased risk estimates and comparatively poor generalization performance[13]. One common source of data leakage is incorrect test set design. The data splitting into training, validation, and test sets has to fit the intended application of the model, with the test set resembling production data. For the use case of personalized medicine, where the model needs to generalize to unseen patients, the test set must not contain cell lines used for training or tuning. We refer to this as the leave-cell-line-out (LCO) setting. In contrast, cancer drug repurposing aims to determine whether a drug effective in one cancer type may also be effective in another. This application scenario requires models to generalize across tissue types. Therefore, the test set must contain tissues of origin not seen during training (leave-tissue-out/LTO). In principle, the global models could be applied to drug design. Here, the model must generalize over the chemical input space, and the test set should

contain only unseen drugs (leave-drug-out/LDO). Leave-random-drug-cell-line-pairs-out (LPO) is only warranted when the goal is to evaluate the ability to impute missing values, e.g., for models that guide treatment selection based on a few measured drug responses. Previous studies have shown that models especially struggle in the LDO setting and that seemingly good performance measures for LPO and LCO can be reached by predicting the average response of each drug across the training cell lines[8,14]. Developers should, hence, explicitly state for which application scenarios their model is developed and then evaluate its performance in the proper setting against the appropriate baselines.

3. Effective Sample Size and Pseudoreplication: Drug response datasets provide an extensive number of measurements (Fig. 1, Supplementary Table 2, Supplementary Fig. 8), for example, about 400 000 drug response curves for CTRPv2. This apparent abundance of data prompts developers to implement complex deep learning models with millions of learnable parameters[8]. However, these data points are not independent but stem from a limited set of biological and chemical entities. For CTRPv2, all responses were measured in 886 unique cell lines and 545 compounds. Thus, when aiming to generalize to an unseen cell line or drug, the model has effectively only been exposed to a small subset of the biological or chemical space, leading to sparse and potentially biased coverage. Hence, while effective sample sizes are still that low, complex machine learning models are likely to overfit. Furthermore, statistical testing of model performances can be misdesigned due to pseudoreplication. Pseudoreplication refers to treating non-independent observations as if they were independent in statistical analyses[15]. If each drug-cell line pair is counted as an independent sample rather than recognizing the cell line or drug as the true unit of replication and accounting for this grouping in the test, the significance of the results is potentially exaggerated.

4. Biased evaluation: Drugs differ considerably in their pharmacologic mode of action, binding affinity, and cellular uptake rate, leading to activity at vastly different concentrations or doses[14,16]. As a result, the primary source of variance in drug response data, measured by half-maximal concentrations such as IC50, arises from differences in the mean drug response across these diverse mechanisms of action of the anti-cancer drugs. This gives rise to a version of Simpson's paradox during evaluation: a model that simply memorizes each drug's mean IC50 value can explain most of the variance in the data[14]. Further details are provided in the Supplementary Discussion (1.2 Simpson's Paradox in Drug Response Prediction). However, this success is misleading, as the model fails to capture meaningful drug sensitivity patterns driven by the phenotypic differences in cell lines, thus not learning any generalizable biomedical insights. Additionally, tuning workflows must be published alongside the final set of hyperparameters to show that they were not selected based on data dredging (i.e., repeatedly testing different configurations on the test set on which the final metrics are reported until finding ones that yield favorable results)[17]. Furthermore, if the baseline methods are not thoroughly tuned, their performance is likely underestimated, making the new model seem superior.

5. Missing ablation studies: To build on the current state-of-the-art, researchers need to be able to assess which modeling components are currently successful[18]. While drug response models grow in complexity and increasingly incorporate multi-modal data, ablation studies (systematic perturbation of inputs and model parts) are rarely performed[8,9]. Without proper ablation studies, it is difficult to pinpoint the sources of improvement, making the evaluation of the model's contributions incomplete.

6. Inconsistency of data and lack of consistent benchmarks: Drug response data is inconsistently preprocessed: Studies apply different normalization techniques, such as min-max scaling or z-score normalization, to the logarithmic IC50 target, or binarize responses as sensitive or resistant. Given that the IC50 is not distributed bimodally, thresholds for this distinction are often set subjectively. These differences render response metrics (Supplementary Fig. 4 and Supplementary Fig. 9) and reported error scores incomparable and make a standardized benchmark with uniformly processed response data a prerequisite for progress. Scientific fields have seen dramatic leaps in progress fueled by standardized benchmarks. Benchmarks like ImageNet[19], GLUE[20], and CASP[21] provided rigorous, community-accepted evaluation standards that enabled transparent comparisons and drove algorithmic innovation. In contrast, drug response prediction still lacks a robust, shared benchmark.

Several benchmark studies assess the relative effectiveness of drug response modeling strategies, focusing on specific components such as gene embeddings[22], transcriptome-based vs. marker-based models[23], dimensionality reduction[24], pathway-based neural networks[14], and multi-omics integration[25]. Cross-study generalization is examined by Xia et al.[26] and Partin et al.[27], who both find generalization to be limited due to overfitting and assay-specific differences (e.g., Syto60, a fluorescence-based assay, in GDSC1 vs. CellTiter-Glo, an ATP-based assay, in other screens). Efforts to establish fair and consistent benchmarking protocols fall short in key areas. Hauptmann et al.[25] re-implement multi-omics models with standardized pre-processing, hyperparameter tuning, statistical comparisons, and ablation studies. However, complex models are not compared against simple baseline models, and the implementation does not allow straightforward ways to contribute, limiting future extension and broad adoption in the field. Szalai et al.[28] highlight inflated performance due to drug-specific mean IC50 differences and the fully randomized splitting, and propose bias-corrected evaluation metrics. However, their scope is limited to three basic models trained and tested on the GDSC2 dataset, only using gene expression and mutation features as predictors. Implementation of their bias correction is available as a collection of Jupyter notebooks. They further provide predefined data splits. This, however, constrains the reproducibility and testing of more general settings, which hinders the automation of evaluation and comparison workflows by other researchers. Partin et al.[27] introduce a Python-based benchmarking package focused on cross-dataset generalization. While being standardized and allowing automation, it requires separate scripts and re-implementations per model. Further, it still lacks support for biologically meaningful data splitting strategies[29]. No publication examines the generalization to unseen tissues of origins (LTO setting). Besides, drug response data employed by most studies disregard the variability between replicates.

Here, we introduce DrEval, a framework for benchmarking drug response prediction models to address the challenges of the field. Unlike prior studies, which typically focus on specific datasets or evaluation scenarios, DrEval unifies these efforts into a single, standardized, and extensible pipeline. It integrates core functionalities such as unified model interfaces, dataset harmonization, and cross-dataset as well as bias-aware performance evaluation, while also extending beyond previous benchmarks to enable systematic comparisons across model classes and evaluation schemes. DrEval supports realistic splitting scenarios that test model applicability in relevant real-world scenarios. It enables unrestricted and custom hyperparameter tuning, evaluates generalization capabilities, and is available as an accompanying Nextflow pipeline for scalability in high-performance computing environments. Instead of delivering a state-of-the-art snapshot, DrEval is designed as a continuously extensible framework supporting systematic experimentation, facilitating integration of new models and datasets, and promoting long-term

reproducibility and collaboration. By implementing state-of-the-art models alongside a broad set of baselines in DrEval, we demonstrate how common pitfalls in drug response prediction can be systematically identified. Specifically, we show that model predictions rely mostly on memorizing drug-specific response means and struggle to generalize to unseen drugs. We further demonstrate that generalization beyond tissue context or across screens remains a major challenge. Finally, we highlight how DrEval facilitates model development by enabling feature ablation studies and by providing a framework in which existing architectures can be readily adapted to new data modalities. For this work, we use DrEval to perform a comprehensive benchmark of various models for drug response prediction. A detailed comparison to prior benchmarking efforts is provided in Supplementary Data 1, highlighting both the aspects that were empirically benchmarked and those that could, in principle, be evaluated within the frameworks.

## Results

### The DrEval framework

DrEval is designed to be a living benchmark for drug response prediction. We ensure that evaluations are bias-free, application-oriented, and reproducible, allowing researchers to focus on advancing their modeling innovations by automating standardized evaluation protocols and preprocessing workflows. DrEval's flexible model interface supports various architectures, ranging from statistical models (e.g., matrix factorization) to complex neural networks (e.g., Graph-Neural Networks), as long as they are inductive (i.e., only the training features and labels are known during the training stage) and implemented for monotherapy prediction using a response summary metric like IC50 or AUC. While our pre-defined neural network models are implemented in PyTorch, users can integrate models using other frameworks using an optional dependency. Meanwhile, we enforce strict management of the predictor, preventing target leakage and ensuring bias-free evaluation.

DrEval consists of a standalone Python package available on PyPI as `drevalpy` and an accompanying Nextflow pipeline, which is part of nf-core[30], a framework for community-curated bioinformatics pipelines. The pipeline enables executing DrEval on large compute clusters in a reproducible environment, guaranteeing reproducibility and scalability. Users automatically benefit from Nextflow features such as execution reports detailing runtime and memory usage. Further, the nf-core initiative promotes collaboration over redundancy by encouraging researchers to jointly develop and maintain best-practice pipelines rather than independently re-implementing similar solutions. In the context of drug response prediction, this consolidates efforts into a single, well-maintained pipeline, reducing fragmentation, fostering reproducibility, and providing a straightforward, standardized workflow that serves as a guideline for implementing, evaluating, and sharing drug response models. DrEval adheres to FAIReR standards: datasets and code used to run the models are findable (F) and accessible (A), the pipeline is executable with conda, Docker, or Singularity, hence, interoperable (I), through the pipeline or the standalone, models and evaluations are runnable and reusable (R) via one command, making all our shown results reproducible (eR)[31]. In contrast to existing benchmarks, DrEval fits curves using replicate-level data with CurveCurator[32], to allow for consistent, robust estimation of common viability-based drug response measures across datasets (Supplementary Fig. 4).

To showcase the utility of DrEval, we integrate three published models representing diverse strategies: SRMF (matrix factorization), DIPK (complex deep neural network), and SuperFELTR (single-drug model, multi-omics late integration). We also include baselines of varying complexity: naive predictors, Elastic Net, Random Forest, and neural networks. The neural networks utilize either gene-expression or multi-omics cell line input, alongside different drug representations in the form of fingerprints, ChemBERTa embeddings, or molecule graphs. We evaluate all models across application settings (LPO, LCO, LTO, LDO), training on the largest dataset, CTRPv2, and testing generalization to other public datasets.

### Models learn limited biological signals beyond drug means

An overview of our benchmark results is shown in Fig. 2 (the full results with all metrics and standard errors are listed in Supplementary Tables 5 and 6). About half of the tested models do not significantly outperform the NaiveMeanEffectsPredictor, which solely relies on drug and cell line mean effects (Critical difference diagrams in Supplementary Fig. 5, statistical test details in Supplementary Tables 13–17). In LCO, the setting relevant to personalized medicine applications, no model surpasses a tuned Random Forest (Supplementary Fig. 5b and Supplementary Table 15). Our series of naive predictors reveals how models can appear effective by memorizing dataset-specific effects and allows us to assess the relative impact of these factors on predictive performance (Table 1). Most of the explainable variation in drug response is driven by the drug identity. Smaller contributions stem from cell line biases and tissue identities, which can be learned implicitly by models.

To isolate the predictive signal of differential drug response beyond mean effects, we compute normalized performance metrics by removing the mean drug and cell line effects from both true and predicted responses before metric calculation. In the LCO experiments, DIPK and the Random Forest can only explain 11% and 19% of the differential drug sensitivity (normalized $R^2$), respectively. While DIPK achieves an overall Pearson correlation of 0.91 in LPO, the per-drug correlation drops to 0.56, whereas the per-cell line correlation remains nearly unchanged (0.89). The discrepancy between overall and per-drug performance arises from Simpson's paradox, as illustrated in Fig. 3 for LPO and LCO and Supplementary Fig. 6 for LDO. Supplementary Fig. 1 and Supplementary Discussion 1.2: Simpson's Paradox in Drug Response Prediction provide a more detailed analysis of the issue.

Together, these results show that models capture only a limited fraction of the biologically relevant variation and lack the accuracy required for clinical or translational use. Although DIPK performs comparatively well in the LPO setting, it is the most computationally expensive model. Other models achieve comparable performance with far less training time (Supplementary Table 7). This highlights the importance of benchmarking against simpler models to ensure that increased complexity translates into meaningful performance gains, given the increased resource consumption.

### Models do not generalize to unseen drugs

For the LDO setting, where the task is to predict responses for unseen drugs, no model significantly outperforms the NaiveMeanEffectsPredictor (Fig. 4, statistical test details in Supplementary Table 17). The near-zero $R^2$ values indicate that only a negligible amount of the response variance is explained (Fig. 2). The tested models fail to learn the relation between molecular drug features (MolGNet encodings for DIPK, Morgan fingerprints for all other models) and pharmacological effects. They do not generalize over the chemical space, likely due to the limited number of drugs in the screens and the high complexity of the structure-activity relationship. For a more detailed analysis, including a drug feature ablation study with additional drug encodings (Supplementary Table 1), see Supplementary Discussion 1.3: Difficulties in Extending Predictions to New Drugs with Supplementary Figs. 2 and 3.

### Weak generalization beyond tissue context

We evaluate the models' ability to generalize to an unseen tissue of origin in the LTO setting. LTO reflects a model's ability to transfer mechanistic signals across biological contexts, instead of relying on

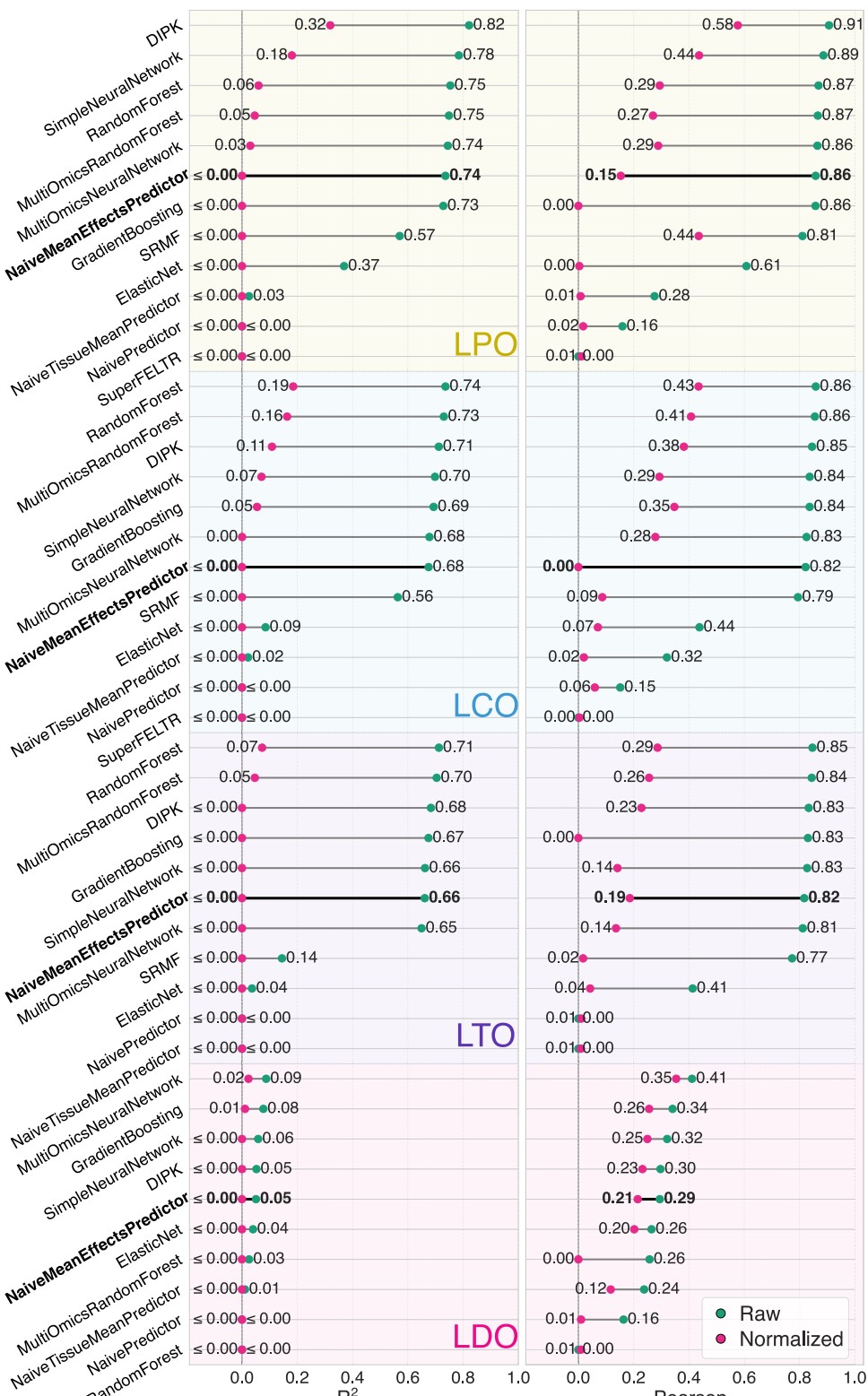

**Fig. 2 | Comparison of model performance for the different test set designs.** LPO: leave-random-drug-cell-line-pairs-out (yellow background), LCO: leave-cell-line-out (blue background), LTO: leave-tissue-out (purple background), LDO: leave-drug-out (pink background). We report the mean $R^2$ and Pearson's correlation (green) between predicted and ground truth ln$IC$50 over the $k = 10$ cross-validation folds and normalized $R^2$ and normalized Pearson's correlation (pink). The normalized metrics are derived by subtracting the predictions of the NaiveMeanEffectsPredictor from the true and predicted values and then recalculating the $R^2$ or Pearson's correlation. There were approximately $n = 23\,600$ points per fold with slight variations depending on the evaluation setting. The bold NaiveMeanEffectsPredictor represents the best performance without utilizing any features, by exploiting cell line and drug biases. SRMF and SuperFELTR are omitted from LDO as they can not predict unseen drugs. Negative (normalized) $R^2$ values are clipped at zero.

tissue-specific mean differences seen during training. In the unnormalized performance metrics, only a small decrease is observed compared to the LCO setting, as these metrics are primarily dominated by the drug-specific effects. However, when using normalized metrics, we observe a substantial drop in performance (e.g., Random Forest Pearson correlation from 0.43 to 0.29). This indicates that the LCO models have implicitly modeled tissue-identities and that robust generalization to unseen tissues (required for drug repurposing) remains challenging, even when the model hyperparameters are tuned towards it.

### Weak generalization across datasets

By applying the models trained on CTRPv2 to predict the unseen responses of CTRPv1 and CCLE, we assess the cross-study generalization of the models in a setting where the molecular input data originate from the same profiling screens, so that the only systematic differences stem from the drug sensitivity assays. Cross-study prediction performances are considerably lower compared to within-study performance (Fig. 5 and Supplementary Table 8). These observations are consistent over different response metrics (Supplementary Table 9: comparison ln$IC50$, AUC, pEC50). The results indicate that, despite harmonizing the response data using CurveCurator, substantial technical differences between the assays persist, causing weak generalization across datasets. Notably, decision tree-based ensemble

models (GradientBoosting, RandomForest) outperform all other model types in cross-study scenarios.

Due to the limited overlap between RNAseq and RRBS methylation data with GDSC1 and GDSC2, these datasets are typically paired with microarray-based gene expression and BeadChip methylation data (Supplementary Table 3). This represents a substantial distribution shift in the input features (See Supplementary Fig. 7), leading to a further drop in performance when models are applied without accounting for these differences. We additionally investigated model generalization to clinically more realistic data using the BeatAML2[33] and the PDX_Bruna[34] datasets, both of which constitute ex vivo drug sensitivity screens: BeatAML2 derived from acute myeloid leukemia (AML) patients and PDX_Bruna from breast cancer patient-derived tumor xenografts. While ref. 33 provides RNAseq measurements,[34] is profiled using microarrays. Model performance declines even more substantially on these datasets, though some models are able to exceed the Naive Mean Effects Predictor on the PDX_Bruna dataset.

### Validating model component utility requires ablation studies

We conduct an ablation study to assess the contribution of each data modality by randomly permuting or invariantly randomizing all features of an individual data modality (Fig. 6 and Supplementary Table 10). We chose DIPK since it was the best-performing multi-modal model and a Multi-Omics Random Forest as a baseline. DIPK relies on gene expression and BIONIC features for cell line representation, and MolGNet encodings for drug representation. The Multi-omics Random Forest is an extension of the gene expression-based Random Forest that additionally incorporates copy number variation, methylation, and mutation data.

Only gene expression and fingerprint/MolGNet randomization result in considerable performance drops. In the LPO setting, both inputs contribute equally to performance, while in the LCO setting, permutations of the gene expression input have a greater impact than permuting the drug input. The drug feature permutation causes the strongest performance drop in the LDO setting, where the model relies on molecular features to extrapolate to unseen drugs. In the LPO and LCO settings, model performance is more dependent on capturing cell line characteristics, which are better represented by gene expression profiles. The Multi-omics Random Forest did not recover any additional predictive signal from other omics modalities, as randomizing them did not result in a considerable performance drop. For DIPK, the drug means, and the cell-line specific effects extracted from the gene expression, sufficed to reach good performance scores in the LPO and

**Table 1 | Mean $R^2$ ± standard error across cross-validation folds for naive mean predictors, trained in the LPO setting on CTRPv2**

| Predictor | $R^2$ |
|---|---|
| Drug and Cell Line Effects | 0.736 ± 0.002 |
| Drug Mean | 0.675 ± 0.002 |
| Cell Line Mean | 0.060 ± 0.001 |
| Tissue Mean | 0.025 ± 0.001 |
| Overall Mean | 0.000 ± 0.000 |

LPO refers to a random split; drugs and cell lines that appear in the test set can also occur in the training set. $R^2$ was calculated over all true versus predicted ln$IC50$ values for each of the $k = 10$ test folds (approximately $n = 23\,600$ points per fold). Overall Mean: NaivePredictor; predicts the training set mean. Drug and Cell Line Effects: NaiveMeanEffectsPredictor; defined in Equation (1). Predicting the mean training ln$IC50$ per drug (Drug Mean: NaiveDrugMeanPredictor) for a test drug-cell-line combination already results in 67.5% explained variance, showcasing how Simpson's paradox distorts global metrics.

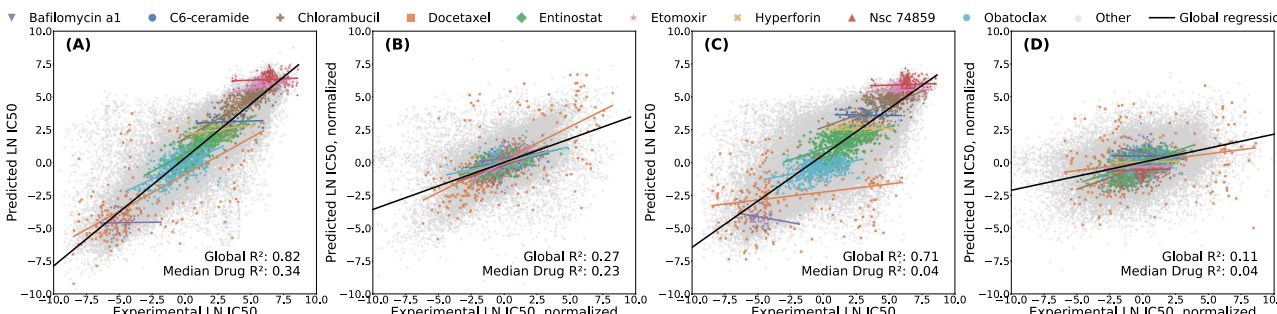

**Fig. 3 | Simpson's paradox in drug response prediction.** Values for nine example drugs are shown in different colors and shapes; the global regression line is shown in black. **A** Predicted ln$IC50$ vs. ground truth for the DIPK model under leave-random-drug-cell-line-pairs-out (LPO) cross-validation. The apparent correlation is largely driven by differences in mean drug potency. **B** After subtracting drug and cell line mean effects for the LPO results, only a weak signal remains, indicating limited learning of differential response beyond remembering mean cell line and drug responses. **C** Non-normalized coefficients of determination are lower under leave-cell-line-out (LCO) cross-validation. **D** After subtracting drug means for the LCO results, a minimal predictive signal remains. (**A**) and (**C**) show how Simpson's paradox distorts the global performance metrics as the global $R^2$ is much higher than the $R^2$'s per drug, creating a misleading image of the model's performance. LDO setting shown in Supplementary Fig. 6. The global $R^2$'s have been calculated over ~ 220 000 data points. The number of points underlying the per-drug $R^2$'s varies between 3 and 793 (median: 430).

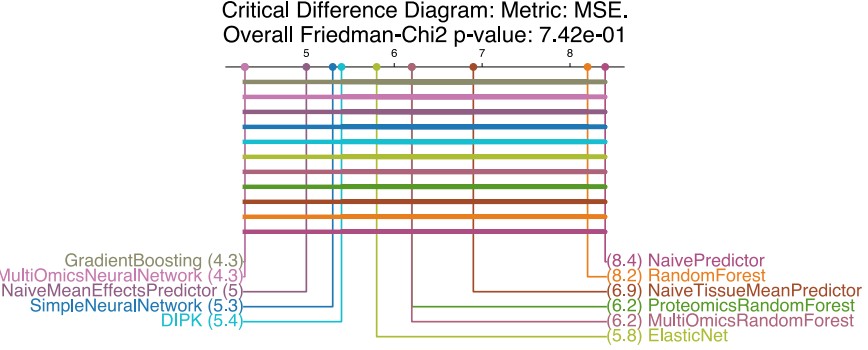

**Fig. 4 | Critical difference diagram for the leave-drug-out (LDO) setting using an MSE-based ranking in the cross-validation folds.** The number in parentheses indicates the mean rank. Overall differences are first assessed using a Friedman test ($p$-value: 0.742): number of tested models as treatments ($t = 11$) and 10 cross-validation folds as blocks ($n = 10$). The Friedman test assumes paired rankings across folds, without distributional assumptions. Then, the diagrams are calculated using an MSE-based ranking in the cross-validation folds. For each model, we draw a horizontal bar, connecting it to all other models from which it does not differ significantly. We assess this using a two-sided pairwise Conover test (Benjamini–Hochberg adjusted $p$-values $< 0.05$) applied to the same paired observations and introducing no additional assumptions. In the LDO setting, no model performs significantly better than any other. Leave-random-drug-cell-line-pairs-out (LPO), leave-cell-line-out (LCO), and leave-tissue-out (LTO) diagrams in Supplementary Fig. 5. Statistical test details are provided in Supplementary Tables 13 (Friedman-Chi$^2$ tests) and 15-18 (pairwise poshoc-Conover tests LPO, LCO, LTO, LDO).

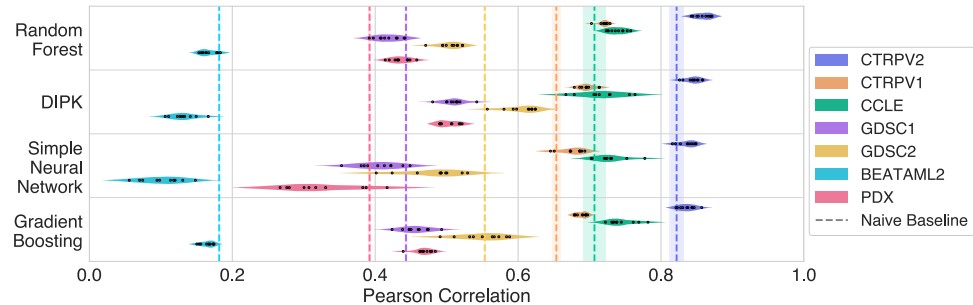

**Fig. 5 | Pearson Correlation of model predictions vs. ground truth ln*IC*50 values for various models.** All models are trained on CTRPv2 and tested on unseen cell lines (LCO). Test sets include the test folds of CTRPv2 (blue, approximately $n_0 = 23\,600$ data points per fold), CTRPv1 (orange, approximately $n_1 = 7500$ points), CCLE (green, approximately $n_2 = 1400$ points), GDSC1 (purple, approximately $n_3 = 74\,000$ points), and GDSC2 (yellow, approximately $n_4 = 42\,000$ points). The trained models further predict the ex vivo responses of the BeatAML2 (cyan, approximately $n_5 = 22\,500$ points) and the PDX_Bruna (pink, approximately $n_6 = 1500$ points) dataset. Drug-cell line combination counts differ from those in Fig. 1 for two reasons: LN_IC50 values deemed unrealistic were set to NA during preprocessing, and some cell lines overlapped with the CTRPv2 training fold. Each correlation value corresponds to a single fold of the cross-validation, computed globally over the $n_0, \ldots, n_6$ test points, respectively. Dashed lines and error bands are the mean $\pm$ SD values of the NaiveMeanEffectPredictor, computed over the $k = 10$ folds, colored by the respective test dataset. Shading represents the standard deviation across cross-validation folds.

LCO setting, while the BIONIC module did not contribute to the predictive power.

These findings highlight the importance of ablation studies in determining whether integrating additional features or model components provides any benefit. Without careful assessment, added features may introduce noise and exacerbate the curse of dimensionality, especially due to the previously mentioned issue of pseudoreplication.

### Community contribution and extensibility: integrating proteomics data

To encourage community-driven benchmarking and reduce duplication of effort, DrEval is designed to be easily extendable. We showcase this by implementing a Random Forest that takes proteomics instead of gene expression data as cell line feature input. Only a few lines of code need to be adapted because we implement a proteomics-specific normalization and change the input to protein expression data. The model can immediately be run using the standalone Python package. The performance of the proteomics-based model is comparable to the gene expression-based version (Supplementary Table 11). Researchers can also integrate entirely new model architectures into DrEval with minimal code, facilitating rapid testing and model design under standardized, reproducible conditions. For a guide with more examples, see https://drevalpy.readthedocs.io/en/latest/runyourmodel.html. If contributed via pull request and included in a new release, it becomes available in the Nextflow pipeline, supporting scalable evaluation on large compute clusters.

### Discussion

Although many cancer drug response prediction models have been proposed, their translation into routine clinical practice or drug development pipelines has so far been very limited[9]. This lack of progress stems from several persistent problems, which DrEval explicitly targets.

First, many models are released without attention to reproducibility, testing, and documentation[12]. A model that cannot be used or retrained easily is less likely to be adopted by the community. DrEval provides a fully versioned Python package and an nf-core pipeline. These come with automated testing, modular design, and continuous integration, ensuring both computational and general reproducibility, as defined by ref. [11]: results generated with DrEval are well reproducible through one command-line call (computational reproducibility) and, because of our various efforts to reduce data leakage and make biases visible, correctly generated (general reproducibility). We

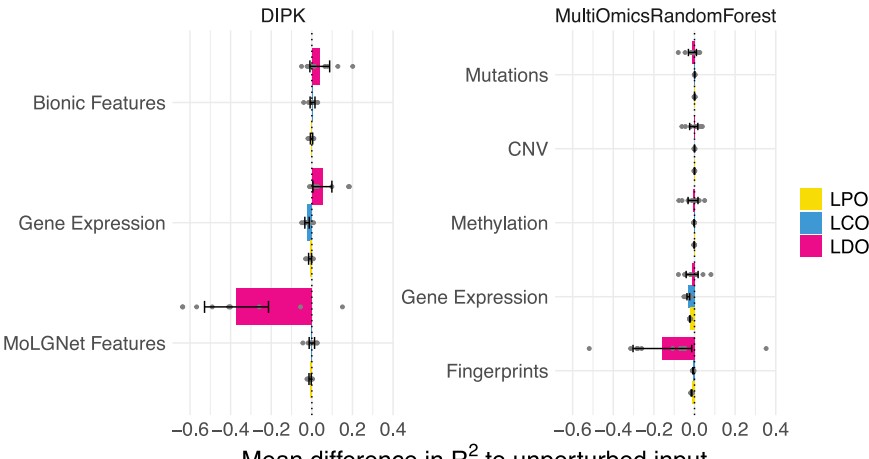

**Fig. 6 | Mean ablation study results of the Multi-omics Random Forest and DIPK.** Results for the leave-random-drug-cell-line-pairs-out (LPO) are shown in yellow, leave-cell-line-out (LCO) in blue, and leave-drug-out (LDO) in pink. One omics input is randomized at a time, such that a drug/cell line receives a random feature of another drug/cell line. Models are retrained from scratch with the randomized features. The $R^2$ of the unperturbed models is subtracted from the $R^2$ of the ablated models for each of the $k = 10$ cross-validation folds, and the results are then averaged: negative values indicate that the perturbation decreased model performance. The number of points per fold varies between ~9600 and ~22 000. Error bars are 95% confidence intervals over the 10 cross-validation folds (center: mean difference, error bands $\pm 1.96 \times$ SEM).

implement a broad range of approaches for comparisons and make contributing easy. Second, in many cases, improper test set design leads to data leakage, and performance metrics are often inflated by drug and cell line mean effects. DrEval enforces strict, use-case-motivated splitting strategies, with normalized metrics to disentangle meaningful prediction from trivial mean effect memorization and overfitting. Third, datasets may appear large due to repeated measurements of the same cell lines and drugs; however, this pseudo-replication inflates the apparent sample size and can mislead model design toward complex, deep learning models. DrEval includes straightforward statistical and ML baselines, making it clear when added complexity yields meaningful gains. Fourth, DrEval uses nested cross-validation with tuning for all models and simple baselines, to allow users to run fair and unbiased comparisons. Fifth, ablation studies are essential for identifying which components contribute to performance as models become increasingly multimodal. DrEval provides built-in standardized support to compare the importance of input features. By implementing a simplified parent model class, users can also benchmark architectural model variants against each other. Finally, inconsistent preprocessing of drug response data across studies leads to distributional shifts that hinder cross-study training. DrEval addresses this by providing standardized and improved preprocessing workflows and harmonized datasets. This enables research towards generalizability.

Using our pipeline, we find that, currently, complex models barely outperform a naive algorithm predicting the mean drug and cell line effects. In line with this, we demonstrate how commonly used metrics are distorted through Simpson's paradox. Simple tree-based models perform best in the LCO setting, which is relevant for personalized medicine. This aligns with previous findings[8,28]. All models explain less than 20% of the drug sensitivity variance when predicting unseen cell lines ($R^2_{normalized} < 0.20$). This performance drops further when predicting cell lines of unseen tissues of origin (LTO, $R^2_{normalized} < 0.08$), a setting relevant for cancer drug repurposing. All tested models fail to predict the effect of an unseen drug (LDO, $R^2 \approx 0$). Further discussion on this issue can be found in the Supplementary Discussion (1.3 Difficulties in Extending Predictions to New Drugs) and in refs. 35–37. Our ablation studies indicate that integrating additional modalities and more advanced gene or drug representations has little effect for the DIPK model, with gene expression being the primary source of

predictive signals in LPO and LCO. Randomization of the molecular fingerprint features only influences performance in the LDO setting, where the models perform worst. Despite our uniform response data processing, models do not generalize well across datasets. This shows that the technical differences between the screens are substantial (e.g., caused by fluorescence- vs. ATP-based screens or different growth conditions at the experimental sites), complicating the formation of a large, joined dataset more suitable for deep learning. We observe that simple ensemble methods, such as Gradient Boosting and Random-Forest, achieve the lowest mean squared error in our cross-study comparisons (Supplementary Table 8). One possible explanation is that the ensemble nature of these models may average out study-specific noise or that the lower representability reduces overfitting to batch effects. A key goal of drug response prediction models is to generalize to patient data. The poor transferability to clinically more realistic BeatAML2 and PDX_Bruna datasets highlights the substantial gap between model development on cell line resources and their intended application in predicting drug responses for patients, when relying on a direct transfer of models to this more realistic domain (for more, see our Supplementary Discussion 1.1: Cross-study prediction on GDSC1, GDSC2, BeatAML2, and PDX_Bruna). Possible solutions to enhance cross-study generalization include transfer learning, establishing standard assays for future screens and solving the problem of dataset integration computationally. In summary, drug response prediction remains a largely unresolved challenge. Despite high reported correlations in the literature, which are inflated due to flawed evaluation, models fail to generalize in realistic settings. Our work highlights these gaps, and our pipeline offers a concrete path forward through rigorous benchmarking, reproducible evaluation, and fast integration of new ideas.

The DrEval framework has several limitations. Some of these limitations are categorized by the Hallmarks of Predictive Oncology proposed by ref. 38 (see Supplementary Table 12 for our self-assessment). Although DrEval is designed to make the addition of new drug response prediction models straightforward, we currently include only three community models. Integrating existing published models remains challenging because many are not implemented in a modular or reusable manner, requiring substantial effort to adapt code, resolve dependencies, and reconstruct the preprocessing pipeline and hyperparameter configurations. These tasks are

typically straightforward for the original authors but difficult to reproduce by others, as also highlighted by ref. 12. We have implemented a diverse set of commonly used approaches (including multi-drug and single-drug models, matrix factorization, gene expression-based and multi-omics deep learning architectures, drug representations in the form of fingerprints, embeddings, or graph neural networks, gene expression autoencoders, prior knowledge integration, and attention mechanisms), but DrEval still lacks several popular methods, such as transformer architectures. We also do not currently provide the framework for transfer learning across biological models (e.g., from cell lines to patient data), fine-tuning, or drug synergy prediction. Standardized functionalities for uncertainty quantification and model interpretability are also left for future work. Nevertheless, DrEval already allows users to implement such models and to export the corresponding result files and trained models, for downstream analyses of uncertainty or interpretability. Another limitation is that DrEval currently relies solely on grid search for hyperparameter optimization, both in the standalone Python implementation and the Nextflow pipeline. While grid search is feasible for relatively fast models with few tunable parameters (Random Forest, Elastic Net), it becomes inefficient for deep learning models, where the parameter space cannot be sufficiently explored. In the Nextflow implementation, each hyperparameter configuration is executed as a separate process, enabling parallel exploration but introducing substantial overhead for faster models. As a result, some of the conclusions, particularly those concerning deep learning models, may be affected by suboptimal hyperparameter choices. We plan to address these limitations by integrating more efficient optimization frameworks, such as optuna[39] or by adopting modules from the machine learning pipeline nf-core/deepmodeloptim, which is currently still under development. In the future, we also plan to implement a similarity-based drug split, aligning with the idea of the LTO split. With our current LDO split, drugs with highly similar structures could still inflate some test performance results[37]. Furthermore, the current framework includes only two curated datasets, which do not originate from immortalized cell lines (BeatAML2, PDX_Bruna). Although DrEval can process custom datasets, we aim to expand the set of curated datasets to ensure consistency and reproducibility for all users. To better assess clinical relevance, subsequent extensions should incorporate data closer to the application scenario, like organoid data or patient drug responses measured by RECIST (Response Evaluation Criteria in Solid Tumors)[40] criteria. Finally, we have not quantified whether the data causes models to exhibit predictive biases with respect to certain cell line subpopulations (including race/ethnicity, sex, age, or cancer types).

Our study inherits the general limitations of cell line viability-based drug response prediction in cancer cell lines: Predicting drug response from baseline omics is inherently more challenging than doing so with post-treatment molecular profiles, which directly reflects a drug's effects and enables more informed inferences about its mode of action and similarity to other compounds. Another major unresolved issue is the gap between in vitro measurements and in vivo outcomes[8]. Further, even if viability could be predicted accurately, this would not directly translate into valid treatment recommendations. IC50 and EC50 values are not directly comparable across drugs because different drugs operate at different concentrations or doses. Additionally, viability does not account for the relative toxicity to healthy tissue (where a measure related to side effects or general toxicity, like the LD50, is needed, which response screens cannot determine)[41]. Another limitation is dataset size. With only hundreds of drugs and cell lines, current datasets are too small to capture the complex biochemical effects of molecules on diverse cellular systems. This limitation can most likely not be addressed through better modeling alone and calls for larger, standardized screening efforts. We hope that such datasets will first enable decent baseline performance,

after which differences between modeling approaches will become more relevant.

In the future, we aim to continuously integrate new models to ensure DrEval remains aligned with state-of-the-art developments and encourage researchers to develop their models directly within DrEval. This turns the benchmark into a continually evolving resource that tracks meaningful progress in the field. Since DrEval supports all possible input types, it can be used to study data from proteomics or epigenomics studies, biological network data, and drug representations. In line with this, we plan to support classification strategies, with particular attention to how sensitivity labels are defined, as current practices often rely on thresholding strategies that may not reflect biological or clinical relevance. We further plan to extend the framework with built-in transfer learning capabilities, enabling users to pre-train models on large-scale datasets and fine-tune them on smaller, more realistic datasets. Additionally, we plan to include metadata in our evaluations to also account for tissue-, sex-, or age-specific effects.

## Methods
### Model interface and integration in DrEval
DrEval defines a model as a function that maps a cell line and drug pair to a continuous response value (regression). The inputs characterize the cell lines (e.g., baseline omics measurements) and drugs (e.g., fingerprints) and are flexible across models. Each model defines its own feature loading. The preprocessing of the input features is seen as part of the modeling process and can be defined uniquely for each model. To ensure a fair comparison between models, DrEval fixes the output across all compared models. Supported outputs are $\ln IC50$, IC50, EC50, pEC50 (negative $\log_{10}$ of the EC50), and AUC. Additionally, arbitrary measures can be used when providing custom datasets, but at the moment, we only support regression models. If not specified otherwise, we predicted $\ln IC50$ for this study as it is most commonly used by state-of-the-art models[8].

DrEval employs a flexible model interface, supporting any models that admit `train` and `predict` procedures. Developers must implement their feature loading and, optionally, a set of hyperparameters for model tuning. Accordingly, DrEval allows for any modeling strategy, such as statistical models, ML, or algorithmic approaches, and any input feature type (including non-tabular data types like networks or images).

### Implemented baseline and literature models
An overview of all implemented models is provided in Supplementary Data 2. We re-implemented several models from the literature, covering a range of different strategies: SRMF (matrix factorization)[42] and DIPK (complex deep neural network with gene and drug features)[43] are included as global models. We also include single-drug models that are fitted and tuned per drug, respectively, by adapting the classifiers MOLI (most cited simple neural network, multi-omics late integration)[44] and SuperFELT (refinement of MOLI)[45] to regression. Researchers can employ these models as baselines to compare against their work, eliminating the need to tediously re-implement the models from disparate source codes.

Furthermore, we offer global, gene expression- and drug fingerprint-based baselines, such as a fully connected neural network (early integration) and various classical ML models (Elastic Net, Random Forest, Gradient Boosting, Support Vector Regressor). We supply different drug representation approaches for the neural network: ChemBERTaNeuralNetwork, which uses pre-trained ChemBERTa embeddings and DrugGNN, which represents the structure as a graph and processes it with a graph neural network. The neural network and random forest are also available as a multi-omics version (additionally employing copy number variation, methylation, and mutation features). The hyperparameter search grids for all models are listed in Supplementary Table 4. DrEval includes a set of naive predictors based

solely on response statistics to assess whether complex models truly capture interactions or simply exploit marginal effects. These consist of the NaivePredictor, which returns the average response of the training set, the NaiveDrugMeanPredictor, which returns the average response for each drug, and the NaiveCellLineMeanPredictor, which predicts the average response per cell line. The NaiveMeanEffectsPredictor combines both sources of variation and predicts responses as the sum of the overall mean with cell line and drug effects

$$\hat{y}_{ij} = \mu + (\mu_i^c - \mu) + (\mu_j^d - \mu) = \mu_i^c + \mu_j^d - \mu, \qquad (1)$$

where $\mu$ is the NaivePredictor, $\mu_i^c$ is the mean for cell line $i$, and $\mu_j^d$ is the mean for drug $j$. If predictions are made for a drug or cell line that was not observed, we set $\mu_i^c = \mu$ or $\mu_j^d = \mu$, respectively. Finally, the NaiveTissueMeanPredictor predicts the mean response of all training cell lines that belong to the same tissue of origin as the cell line for which predictions are made. Despite their simplicity, these models can appear to perform well when variance in the data is attributable to systematic differences in mean drug response.

### Data splitting and hyperparameter tuning

DrEval employs a k-fold cross-validation schema incorporating an inner holdout validation process for model tuning. An early-stopping-validation set is split from the remaining train set for models employing an early stopping mechanism (DIPK, MOLIR, SuperFELTR, Simple- / MultiOmicsNeuralNetwork). For the LCO, LDO, and LTO settings, we ensure that all splits are disjoint for cell lines, drugs, and tissues, respectively. In the LPO setting, we only ensure that replicate experiments involving the same drug-cell line pair are contained within the same data subset. To facilitate fair comparisons, DrEval tunes all models of a benchmark run by performing grid searches on the supplied hyperparameter lists.

### Benchmark data

**Cell viability data.** Raw dose-response data were downloaded from seven publicly available, commonly used datasets: CCLE (Supplement of ref. [3]), CTRPv1/v2 (CTD² Data Portal), GDSC1/2 (cancerrxgene website), BeatAML2 (Github[46]), and PDX_Bruna (figshare associated with[34]). While CCLE, CTRPv1/v2, and GDSC1/2 measured drug response for immortalized cancer cell lines, BeatAML2 and PDX_Bruna conducted the drug response screens on ex vivo samples. BeatAML2 screened freshly isolated mononuclear cells from AML samples, while Bruna et al. used patient-derived tumor xenograft (PDTX)-derived tumor cells from breast cancer patients. Dataset sizes are shown in Fig. [1] and Supplementary Table 2. All preprocessing scripts are available in the accompanying GitHub repository. All cell line names were mapped to Cellosaurus CVCL accessions, and all drug names were mapped to PubChem CIDs (when available).

To ensure consistency and reduce preprocessing discrepancies, all datasets are reprocessed in a standardized workflow made available via drevalpy (Supplementary Fig. 4). Hence, we also support fitting custom data, facilitating cross-study predictions through compatibility with the provided datasets and future dataset contributions.

Instead of aggregating replicates prior to normalization and curve fitting, as is the standard practice, DrEval includes replicate variability into quality control measures. This source of experimental variability is often overlooked when aggregating replicates prior to fitting, which leads to inaccurate or misleading drug response measures in the case of large discrepancies between replicates. Therefore, raw measurements are normalized per replicate by dividing by the control (no-drug) measurement, yielding viability values from 0 to 1; these are then used by CurveCurator[32] to fit a single model across replicates for each drug-cell line pair. This allows CurveCurator to include residual differences at each measurement point for more robust estimation of quality measures, compared to simply taking the mean or median of

the replicates at each measurement point. For the CCLE dataset, only aggregated data were available, which had already been normalized, so it was treated as a single replicate.

CurveCurator calculates $p$-values using a recalibrated F-statistic for each curve fit, measuring how well the data conforms to a sigmoidal curve, which is the biologically expected phenotypic response to increasing drug dosages. Since different datasets test varying dosage ranges for specific drugs, curve fitting and p-value calculations were conducted within each dosage group to ensure fair comparisons. Finally, EC50 values falling outside the measured dosage range per drug and IC50 values deviating by more than one order of magnitude from the measured range were considered invalid. The datasets can be filtered for quality using the statistical measures provided by CurveCurator, minimizing inter-dataset batch effects. This leaves experimental protocols and the choice of viability assays as the primary potential sources of systematic shifts. If not stated differently, we did not apply any quality filter in our benchmark experiments to maintain comparability to previous studies and avoid data loss.

**Cell line features.** Omics screens corresponding to the employed drug response screens are available from various sources, overlapping at varying proportions (Supplementary Table 3). For the cell line datasets, we provide two sources of gene expression and methylation data because they better complement CCLE/CTRPv1/CTRPv2 or GDSC1/GDSC2. For the CCLE and CTRP screens, multiple gene expression datasets (microarray, RNA-seq) exist with varying measures and preprocessing, leading to inconsistencies. To ensure reproducibility, we reprocessed the raw RNA-seq data from Ghandi et al.[47] (PRJNA523380) using the nf-core RNA-seq pipeline[48] (STAR for alignment[49], Salmon for quantification[50], version 3.10.1). We supply the resulting TPM values. As a second source complementing the GDSC screens, we obtained RMA-normalized microarray expression data from the GDSC Data Portal. Bruna et al.[34] supply processed microarray data in their figshare, which we did not modify. We also did not reprocess the RNAseq data provided by ref. [33] because the FASTQ files were not available. Methylation BeadChip data for GDSC (pre-processed beta values for all CpG islands) were also obtained from the GDSC Data Portal. Further, methylation RRBS data for promoter CpG clusters were downloaded from DepMap (Release: Methylation (RRBS))[29], which has a larger overlap with the CCLE and CTRP screens.

Mutation and copy number variation (CNV) data for the cell line datasets were downloaded from Cell Model Passports[51] and can be combined with all five screens. The mutation data is binary and was filtered to only contain coding mutations that are not silent. CNV is represented as GISTIC scores derived from Affymetrix SNP6.0 array data (integers ranging from − 2: high-level deletion to 2: high-level amplification). We reprocessed the raw copy number alteration data from Cell Model Passports and Bruna et al. with GISTIC2.0[52], available via GenePattern[53]. Proteomics data was obtained from a DIA screen by Gonçalves et al.[54]. The raw data (PRIDE: PXD030304) were filtered for global Q values ≤0.01 and proteotypic peptides. Protein quantities were calculated using the MaxLFQ algorithm of the DIA-NN package (version 1.0.1)[55]. We offer to filter genes with gene sets such as anti-cancer drug target genes curated by GDSC, the landmark genes used by the L1000 assay[56], or user-provided sets.

**Drug features.** We use Morgan fingerprints generated from SMILES with RDKit[57] for drug representation. GDSC1 and GDSC2 already provide SMILES. For the other datasets, they were downloaded from PubChem[58]. DIPK uses 768-dimensional drug encodings generated using the pre-trained graph neural network-based molecule encoder MolGNet[59] and 512-dimensional interactome features computed by averaging the embeddings of the top 256 highly expressed gene embeddings extracted from BIONIC[60], a graph autoencoder trained on multiple gene interaction networks. We adapted DIPK's preprocessing

scripts to generate missing MolGNet features from SMILES. We additionally include ChemBERTa embeddings as an alternative drug representation. Canonical SMILES are tokenized and passed through the pretrained ChemBERTa model[61]. The mean of the final hidden states serves as a molecular embedding with a fixed length of 768. We create graph-based drug features by converting SMILES strings into a graph, where atoms are represented as nodes with categorical attributes for atomic number, degree, formal charge, hydrogen count, hybridization state, aromaticity, and ring membership. The bonds define edges, which have edge features encoding bond type, conjugation, and ring membership.

**Tissue information.** We obtained disease annotation for our LTO setting from Cellosaurus (Release 52) and enriched it with DepMap (22Q2) sample information if Cellosaurus disease information was missing. We then applied a curated tissue synonym dictionary to map specific diseases to broader, biologically meaningful tissue-of-origin categories. Finally, we manually overwrote mappings for misclassified or ambiguous cell lines based on verified external sources (e.g., ATCC, NCI).

### Ablation studies, robustness tests, cross-study prediction

DrEval supports feature ablation via modular input permutations. Models must declare their input feature modalities (e.g., gene expression, mutation), which allows independent permutation of each view. For instance, if a model uses gene expression and mutation data, DrEval fits variants where one modality is permuted across cell lines (e.g., shuffling gene expression profiles), removing any possible connection between this modality and the response. A performance drop indicates the importance of that modality. To determine whether a model relies on informative biological signals rather than simple summary statistics (such as mean and variance), DrEval also implements summary statistics-invariant randomizations. We generate synthetic features drawn from a normal distribution, preserving the original mean and standard deviations for tabular features. For graph inputs, edges are shuffled while preserving node degrees. This shows whether a model learns from specific biological signals or simply fits simple statistical attributes, such as the overall mean gene expression of a cell line. For model component ablations, DrEval does not directly support automated removal or substitution of model architectural elements, since the pipeline only interfaces with model inputs and outputs. Instead, users can implement a modified version of the model without the component of interest and benchmark it within the same evaluation pipeline.

For robustness tests, DrEval refits models with different initializations to evaluate the performance stability. We further implement cross-study predictions to assess cross-dataset generalization performance. In our LPO setting, no cross-study predictions are made for drug-cell line combinations present in the training dataset. In the LCO, LTO and LDO settings, cross-study predictions for cell lines, tissues or drugs found in the training dataset are excluded.

### Evaluation and visualization

We compute the following metrics between the true and predicted responses: mean squared error (MSE), root MSE (RMSE), mean absolute error (MAE), the coefficient of determination ($R^2$), Pearson, Spearman, and Kendall correlation. Metrics are calculated separately for each cross-validation fold based on all test set predictions. In addition, we assess the $R^2$ and correlation metrics stratified by drug and by cell line, aggregating predictions across all folds. We also define normalized $R^2$ and correlation coefficients by subtracting the predictions of the NaiveMeanEffectsPredictor from both predicted and true responses before evaluation. This removes any variance due to average drug and cell line effects. The remaining variance includes both noise and differential drug response, which is the variation driven by

interactions between molecular cell line features and drug properties. These effects are clinically relevant, as they reflect mechanisms of sensitivity and resistance of cancer cells. We compute effect sizes by calculating strictly standardized mean differences for the $R^2$ and MSE. We present the results in an automatically generated HTML report with interactive plots. An example is available on GitHub (https://dilis-lab.github.io/drevalpy-report/). All underlying evaluation metrics, ground truth values, and predictions are also provided as CSV files for further analysis.

### Statistics and reproducibility

We used all available drug-cell line pairs from CTRPv2 (approximately 400 000 response curves across 886 cell lines and 545 compounds) as the primary training and evaluation dataset, as it is the largest publicly available drug sensitivity screen. The effective sample size for generalization is limited by the number of unique biological and chemical entities rather than the number of drug-cell line pairs. The number of cross-validation folds ($k = 10$) was chosen to balance statistical power and computational cost.

Several pre-established exclusion criteria were applied. EC50 values falling outside the measured dosage range for a given drug and IC50 values deviating by more than one order of magnitude from the measured range were considered invalid and excluded. Mutation data were filtered to retain only non-silent coding mutations. Proteomics data were filtered at a global Q value threshold of $\leq 0.01$ and restricted to proteotypic peptides. Cell lines lacking matching omics data were excluded from the corresponding analyses. Due to limited overlap between RNA sequencing and RRBS methylation data in GDSC1 and GDSC2, these datasets were paired with microarray-based gene expression and BeadChip methylation data instead.

In this computational benchmarking study, randomization in the analysis was limited to the assignment of data points to cross-validation folds, which is described in the Data splitting and hyperparameter tuning section. Blinding was not applicable.

We assess statistical significance using the Friedman test on model ranks (ranked by MSE) across CV folds. The Friedman test assumes paired rankings across folds, without distributional assumptions. The test statistic $Q$ has two parameters, $n$ (the number of cross-validation folds) and $k$ (the number of models) and the p-value is calculated using the $\chi^2$ distribution with $k - 1$ degrees of freedom ($P(\chi^2_{k-1} \geq Q)$). Then, a pairwise Conover post-hoc test with Benjamini-Hochberg correction identifies which model pairs differ. The p-value can be derived since the Conover test statistic follows a t-distribution with $(n - 1)(k - 1)$ degrees of freedom ($n$ = number of cross-validation folds, $k$ = number of models). For both tests, we set a significance level of 0.05.

### Benchmarking setup

We benchmark all models using 10-fold cross-validation in the implemented LPO, LCO, LTO, and LDO settings. To ensure fairness, all baseline and test models are tuned based on the validation RMSE using the same cross-validation folds, and performance is assessed using MSE, $R^2$, Pearson correlation, and their normalized variants. The hyperparameter spaces are inferred from the relevant publications or set to balance run time and search space size (Supplementary Table 4). Since it is the largest available dataset, the primary training and evaluation dataset is CTRPv2 with CurveCurator-processed responses. Our cross-study evaluations assess generalization to the CurveCurator-processed responses of CTRPv1 and CCLE using the same input screens. For the generalization to GDSC1 and GDSC2, we use the microarray input instead of the RNAseq input and the BeadChip methylation input instead of the RRBS methylation input. For BeatAML2 and PDX_Bruna, we use their associated gene expression input data.

## Reporting summary

Further information on research design is available in the Nature Portfolio Reporting Summary linked to this article.

## Data availability

The drug response, cell line, and drug data used in this study are available in the Zenodo database under accession code 12633909 (10.5281/zenodo.12633909). Curve fits for the cell line datasets can be explored at ProteomicsDB: https://www.proteomicsdb.org/analytics/cellSensitivity[62]. The data underlying all Figures and Tables are provided in the Source Data file (https://doi.org/10.6084/m9.figshare.30724907).

## Code availability

DrEval consists of a Python package, available on PyPI (`drevalpy`) and GitHub (https://github.com/daisybio/drevalpy), which has been archived on Zenodo under accession code 18302238 (https://doi.org/10.5281/zenodo.18302237, v1.4.1)[63]. The suite also includes an accompanying nf-core pipeline available at https://github.com/nf-core/drugresponseeval (Zenodo 10.5281/zenodo.14779984, v1.2.0)[64]. The preprocessing scripts that lead from the original data to the data provided on Zenodo are available at https://github.com/daisybio/preprocess_drp_data (v1.0.0).

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

## Acknowledgements

J.B., M.P., M.W., and M.L. were supported by the German Federal Ministry of Education and Research (BMBF) within the framework of the CompLS funding concept [031L0305A (DROP2AI)]. Funded by the Deutsche Forschungsgemeinschaft (DFG, German Research Foundation) [422216132]. The results published here are partially based upon data generated by the Cancer Target Discovery and Development (CTD$^2$) Network (https://www.cancer.gov/ccg/research/functional-genomics/ctd2) established by the National Cancer Institute's Center for Cancer Genomics. We thank Jonah Reiner for his help with the integration of the DIPK model. We would also like to thank the nf-core team for welcoming us to the nf-core community and for their assistance in reviewing and releasing the pipeline. We would also like to thank the nf-core community for developing the nf-core infrastructure and resources for Nextflow pipelines. A full list of nf-core community members is available at https://nf-co.re/community.

## Author contributions

J.B. (conceptualization [equal], data curation [equal], formal analysis [equal], investigation [equal], methodology [equal], software [equal], validation [equal], visualization [equal], writing—original draft [equal], writing—review & editing [equal]), P.I. (conceptualization [equal], data curation [equal], formal analysis [equal], investigation [equal], methodology [equal], software [equal], validation [equal], visualization [equal], writing—original draft [equal], writing—review & editing [equal]), M.P. (conceptualization [supporting], data curation [supporting], formal analysis [supporting], software [supporting], writing—original draft [supporting], writing—review & editing [equal]), M.W. (conceptualization [supporting], funding acquisition [equal], resources [equal], supervision [supporting], writing—review & editing [equal]), K.B. (conceptualization [equal], funding acquisition [equal], project administration [equal], resources [equal], supervision [equal], writing—review & editing [equal]), M.L. (conceptualization [equal], funding acquisition [equal], project administration [equal], resources [equal], supervision [equal], writing—review & editing [equal]).

## Funding

## Competing interests

M.L. consults for mbiomics GmbH. M.W. is a founder and shareholder of MSAID GmbH with no operational role in the company. All other authors declare no competing interests.
