## [Transparent Peer Review file · Nature Communications]

Critical Evaluation of Drug Response Prediction Models with DrEval

Corresponding Author: Professor Markus List

Version 0:

Reviewer comments:

Reviewer #1

(Remarks to the Author)
DrEval

The paper introduces DrEval, a reproducible benchmarking framework for drug-response prediction. It discusses an important topic of the reproducibility crisis in drug response prediction field and attempts to alleviate situation, which is very beneficial to the research community.

Major comments:

1. Table S2 suggests that cell lines number in the DrEval sets significantly differs from the cell line number in original datasets (e.g., CCLE). It suggests that there are some exclusion criteria that are not well-described in paper which is briefly mentioned in section 4.4. However, as large portions of the dataset were discarded, it would be interesting to understand all data filtering steps in detail (probably, in supplementary material). Current filtering level seems quite stringent and may not necessarily be a good assessment for the drug response prediction models on the real-world data. I suggest that authors to explain each step to the readers in details and allow to manually adjust filtering level.
2. Various statistics of the curated datasets are also of interest - total number of drugs, total number of experiments, response value distribution, etc.
3. Although CurveCurator harmonises viability curves, differences in assay type (ATP- vs fluorescence-based) and growth conditions may still be sources of bias.
4. Ablation studies infrastructure is partially present; however, the authors tend to conflate it with hyperparameter tuning. The authors provide an interface for developing new models that can ingest hyperparameters from the input files. This is only a portion of the complexity management for the ablation studies framework. First, it provides users with very limited support for another widespread (especially, for the DL) ablation study type - changes in the model's architecture. The user who develops a new model with DrEval help still has to manage all internal complexity of the model relevant to the ablation study. Second, and most importantly, hyperparameter tuning is computationally expensive. I was unable to find the code responsible for the actual tuning strategies (e.g., grid search) in the framework repository. The supplementary data also suggests that the authors ran a very limited hyperparameter search experiment, predominantly on simpler models. I am not sure that the current version of DrEval supports hyperparameter search scaling to a reasonable search space for the modern DL-based drug response prediction models. I compel authors to more clearly define each part of this section with regard to the application, level of support, and supported scale of this part of DrEval framework.
5. Table 1 is not clearly explained. Are cross-validations folds in LPO setting it refers to random (e.g., standard k-fold CV), or are they different cross validations performed over different group-based CVs (e.g., leave group out CV for tissue)? Do the results represent the average value of R^2 for each group in 'Naive Predictor' column for the data predicted on the k different test sets of the k-fold CV that cover the entire dataset? 'Overall Mean' is not defined.
6. Statistical tests in 4.6 (Friedman, Conover) are performed multiple times. Multiple hypothesis testing correction is not discussed.
7. Literature overview and comparison with the existing work are rather limited. Regarding the splits, some work (e.g., <https://arxiv.org/pdf/2406.00873>) discuss an additional alternative strategy. Conducting a more comprehensive literature review would help authors to better position their work in the field.

(Remarks on code availability)

1. The experiment interface currently relies on a large number of 'Metric' variables in the drug_response_experiment, which is confusing. Code refers to it as 'metric for tuning hyperparameters' while not defining target metric for the response prediction.
2. Randomization mode in the code iterates over different data modalities for the same model. As data tables may have different dimensionality and input types (e.g., expression data, mutations), it is extremely unlikely that a user would need to manually iterate over them for a single module.
3. Better guidance for the model construction/analysis with the framework would be helpful.

Reviewer #2

(Remarks to the Author)

Summary

This paper tackles an important issue in drug response prediction (DRP): the need for standardized and reproducible evaluation of DRP models. The authors present DrEval, a Python-based benchmarking framework designed to address common challenges in the field, such as inconsistent data handling, lack of ablation studies, and data leakage. The framework is tested on three community models, two deep learning-based and one non-DL, and the study includes cross-dataset comparisons.

That said, several aspects of the manuscript would benefit from more clarity. In particular, the comparison with prior benchmarking studies often blurs the line between what this study actually demonstrates and what the proposed framework could, in theory, support. It's important to separate these points to clearly define the contribution. Also, while hyperparameter tuning is emphasized throughout, the actual search space for many models looks quite limited. That raises questions about how robust the comparisons are. Lastly, the limitations section focuses mostly on general challenges in the field, rather than reflecting on limitations specific to this study or framework.

The comments below aim to help clarify the contribution and strengthen the overall presentation.

Major comments

1. Comparison with existing benchmarking studies

Section 3.1 outlines limitations of prior benchmarking studies and highlights the added value of the current work. However, the discussion often blends two distinct dimensions:

- a. Improvements reflected in the analyses actually performed in this study.
- b. Improvements offered by the extensibility of the framework, even if not yet demonstrated.

I suggest clearly distinguishing these dimensions when drawing comparisons. For example:

- a. "Hauptmann and Kramer [2023] reimplement multi-omics models with standardized preprocessing, hyperparameter tuning, statistical comparisons, and ablation studies. However, their evaluation focuses on classification tasks and excludes the relevant LCO, LTO, and LDO prediction scenarios." This appears to reflect a difference in study scope, rather than limitations of their framework. By the same token, one could say this study focuses on regression, while Hauptmann and Kramer focused on classification.
- b. "Partin et al. [2025] introduce a Python-based benchmarking package focused on cross-dataset generalization ... lacks support for biologically meaningful data splitting ... only uses omics input from DepMap ... uses a narrow hyperparameter space." It's unclear whether these are limitations of the framework itself or choices made in the scope of the accompanying study. For instance, could their framework support other data types or splitting strategies?

Additionally, some of the cited works (e.g., Nguyen 2017 on dimensionality reduction, Li 2023 on pathway-based neural networks) are mentioned briefly but not integrated into the comparative discussion. Clarifying whether these types of analyses are supported or comparable in the proposed framework would strengthen the narrative.

To make the comparisons more transparent, consider including a table contrasting this work with others along key dimensions, such as:

- a. Scope and scale of analysis: e.g., LTO/LDO/LCO/LPO, cross-study generalization, interpretability.
- b. Model types: Distinguish between community models (e.g., MOLI, SRMF, DIPK) and simpler baselines (e.g., RF, GB). You might also consider a DL vs. ML distinction, and naïve vs. other baselines.
- c. Automation: e.g., hyperparameter tuning, pipeline orchestration.
- d. Extensibility: e.g., ease of integrating new community models, datasets, evaluation metrics, DL frameworks (PyTorch, Keras), or external biological data (e.g., pathways).

2. Hyperparameter tuning

Throughout the manuscript, the authors emphasize the importance of robust hyperparameter optimization (HPO) and claim that their framework supports and applies consistent HPO practices across all models. A few representative statements:

- “if the baseline methods are not thoroughly tuned, their performance is likely underestimated...” (Page 3, line 124)
- “DrEval includes fair and consistent hyperparameter tuning.” (Page 3, line 160)
- “DrEval uses nested cross-validation with tuning for all models and simple baselines...” (Page 7, line 417)
- “The hyperparameter search grids for all models are listed in Table S3.” (Page 8, line 537)
- “We enforce a fair setting by tuning all baselines...” (Page 8, line 566)

The authors also critique other works for lacking robust HPO, e.g.:

- “Many works ... do not publish a tuning workflow.” (Page 3, line 118)
- “Partin et al. [2025] ... evaluates the provided models in a narrow hyperparameter space.” (Page 4, line 218)

However, based on Tables S2 and S3, the actual hyperparameter (HP) search for many baseline models (S2) and community models (S3) appears quite limited. For several models, important HPs (e.g., learning rate, batch size, dropout rate) were not tuned at all. In some cases, the authors report using fixed configurations from original papers without any additional tuning. While the authors justify this as a way to avoid overfitting due to undocumented tuning procedures (“prevents possible further implicit data dredging...”), this seems inconsistent with the strong claims in the main text about performing robust and consistent hyperparameter optimization.

This raises two concerns:

- a. The claim that robust HPO was performed across all models, along with criticism of other studies, may be overstated and would benefit from moderation.
- b. Some conclusions, especially those comparing models, may be affected by the limited HPO. If the authors believe the results would hold under a more thorough tuning strategy, an explanation should be provided.

3. Limitations section:

The current discussion on study limitations (Discussion section) primarily restates known limitations of preclinical drug response studies. These are valid concerns, but largely reflect field-wide limitations, rather than limitations specific to the scope and methods in the current study.

I recommend the authors reflect more critically on study-specific and framework-specific limitations. One helpful resource could be the “Hallmarks of Predictive Oncology” framework proposed by Singhal et al. (2025), which outlines seven hallmarks by which predictive oncology models can be assessed: Data Relevance and Actionability, Expressive Architecture, Standardized Benchmarking, Demonstrated Generalizability, Mechanistic Interpretability, Accessibility and Reproducibility, Fairness. The authors might consider identifying which hallmarks are addressed in the study, which are supported (but not demonstrated), and which remain out of scope for various reasons. This could also help highlight areas for future work.

Here are a few areas to consider:

- Scope of analysis: Are there important modeling settings not yet supported? E.g., drug combination or synergy prediction, transfer learning across biological models (e.g., from cell lines to PDX or patient tumors), support for uncertainty quantification.
- Hyperparameter tuning: As mentioned earlier, the tuning strategy (HPs and their ranges) is limited, which may affect some conclusions.
- Interpretability: Was any model interpretation conducted (e.g., SHAP, LIME, or biological pathway-level insights)? If not, does the framework support this.
- Framework: Are there current limitations in the modularity and extensibility of the framework (e.g., support for non-tabular data types like imaging data)?

4. Emphasis on translation to clinic and the lack of models predicting on non-cell line data

The manuscript highlights the importance of building models that can eventually translate into clinical settings (Section 1 and 1.1), and claims that the framework makes it easy to add models for comparison (e.g., “We implement state-of-the-art models for comparisons and make contributing easy.”, Page 7, line 403). However, all models currently included appear to be trained and evaluated on cell line data only.

Given the emphasis on clinical relevance, I suggest including at least one model that operates beyond cell line data, such as patient-derived xenografts (PDX), organoids, or patient data. Several published models demonstrate such capabilities:

- <https://www.nature.com/articles/s43018-020-00169-2>
- <https://www.nature.com/articles/s42256-021-00408-w>
- <https://www.biorxiv.org/content/10.1101/2020.11.17.385757v1>

If the authors choose not to add such models, I suggest softening the claim that contributing is “easy.” Also, if contributing is truly straightforward, it’s unclear why only three community models have been included. A brief explanation would be helpful.

5. Temper Overconfident Language

Some statements in the manuscript use strong or absolute language that isn't fully supported by the results or citations. I recommend reviewing the text to make sure claims are clearly scoped, supported with evidence, or softened where needed. A few examples:

Page 3 (line 162): "Its flexible model interface supports any model type ..."

This is too broad. Only three community models are included, all using cell line data and PyTorch. Either clarify what's meant by "any model type" or soften the claim. You could also consider including more diverse models (e.g., transfer learning, synergy prediction, other DL frameworks) to support this claim

Page 7 (line 343): "... none have been translated into use in clinical practice or drug development."

This claim might be inaccurate. Several predictive models are in use in preclinical research, though often as proprietary tools (in industry). Consider softening.

Page 7 (line 398): "... will not be applied or refined by the community."

Too definitive. Suggest something like "is less likely to be adopted."

Page 7 (line 403): "...make contributing easy."

This has already been raised above. If adding models is truly easy, it's worth explaining why only three have been added.

Page 7 (line 401): "... ensuring both computational and general reproducibility."

The distinction between "computational" and "general" reproducibility isn't clear. Consider clarifying how each is defined and demonstrated in this study.

Minor comments

Line 77: "Previous experiments have shown ..."

Consider: "Previous studies have shown ..."

Line 85: In section Pseudoreplication, provide a more clear definition for pseudoreplication. It's not immediately clear what's the "unit" refer to in line 93.

Line 105: Can you provide citation(s) for this? "Drugs differ considerably in their pharmacologic mode of action, binding affinity, and cellular uptake rate, leading to activity at vastly different concentrations or doses"

The text references Simpson's paradox in several places. Consider adding a small illustrative example (e.g., with synthetic data) in the supplement to clarify its practical relevance.

(Remarks on code availability)

The code appears well-organized and actively maintained, with a clear directory structure. However, I haven't tried to run the analysis, so I cannot comment on potential installation issues or reproducibility challenges.

Reviewer #3

(Remarks to the Author)

The manuscript entitled "From Hype to Health Check: Critical Evaluation of Drug Response Prediction Models with DrEval" by Bernett et al. This work presents a comprehensive benchmarking framework, DrEval, to address critical issues in drug response prediction (DRP), including reproducibility, data leakage, biased evaluation, and lack of standardized benchmarks. The authors also provide an extensive evaluation of multiple models across various realistic splitting strategies and datasets. The methodology is one of the strengths of the paper. The use of clear splitting strategies, nested cross-validation, normalized metrics, and initial ablation studies all reflect careful experimental design and contributes positively to the study.

Major comments:

1. This study has some noteworthy results in demonstrating that:

- 1) Most advanced DRP models still perform worse than a simple baseline that only uses the average effects of drugs and cell lines.
- 2) Complex models (such as DIPK) do not show clear performance improvements over well-tuned tree-based models (like Random Forest) in important cases such as Leave-Cell-Line-Out (LCO).
- 3) When using normalized performance metrics, it becomes clear that these models capture very little true biological signal and mostly rely on dataset-specific patterns.
- 4) Models also struggle to generalize across different datasets, likely due to differences in technical setups and assay conditions.

These results are important and suggest that current modeling approaches in this area should be carefully re-examined.

2. The central conclusion that current models are overly optimistic and fail to generalize is largely supported. However, some claims are potentially overstated or require more nuanced support:

1) The "Why" is Under-explored:

The paper clearly shows what is going wrong (the models don't work well), but does not explain why very clearly. The authors suggest that the failure in LDO is because there are "too few drugs" and the "structure-activity relationship is complex." But this is more of a guess than a solid explanation. A deeper analysis is needed. For example: Is the problem caused by weak drug features (like Morgan fingerprints or MolGNet)? Or do the models simply not learn well from these features? Or is there no strong signal in the data to begin with? The ablation studies (Fig. 6, Table S9) are a good step in this direction, but the paper does not discuss them enough when talking about the LDO failure.

2) Simple Models Perform Surprisingly Well:

Table S7 shows that a simple GradientBoosting model often gets the best cross-study MSE. This is very interesting and goes against the idea that we need very complex models. This result should be mentioned and discussed more in the main text. Why might simpler models handle batch effects better? This is an important question that the authors could explore more.

3) Pseudoreplication and Overfitting Connection is Unclear:

The section on pseudoreplication does not clearly explain how it causes overfitting. The paper mentions both terms, but only says that pseudoreplication "makes it harder to choose the right number of features." This is vague and doesn't explain the real reason or show a clear link between the two ideas. The authors should describe this connection more clearly and in more detail.

4) Critiques of Prior Work Lack Evidence:

The paper criticizes earlier studies but does not give enough evidence to support these points. For example, saying that other studies did data dredging is a strong accusation and needs proof. If there is no direct evidence, the paper should focus on clear problems, like the lack of transparent tuning procedures, instead of guessing what other researchers did. Other claims about missing ablation studies are also made without showing specific examples. These points should be supported by data or clearly reworded.

3. I did not find any major issues in the main data analysis or conclusions. The benchmark is well-designed and carefully carried out. The main finding that current complex models do not perform better than strong classical baselines and generalize poorly is well supported by the evidence. However, some results could be explained more thoroughly, as mentioned above. Moreover, to further explore the boundaries of what is achievable, it would be valuable to consider including a broader range of modern deep learning architectures in the benchmark, such as transformer-based models, graph neural networks, large-scale pre-trained models and so on. The natural question arising from this work is: what kind of innovation, if any, will finally consistently surpass the strong baseline of a tuned Random Forest? I wonder about the performance of the latest methods on DRP.

Minor comments

1. CurveCurater panel: The diagram does not visually convey that missing curves are filled and dose ranges are aligned; x- and y-axes lack labels.
2. Ablation schematic: Curves on the right are unlabeled, and no reference line indicates the full-model performance, so the effect of each ablation is unclear.
3. HTML report snapshot: The image is too small to discern table contents or report structure; no caption describes the displayed elements.
4. Pseudocode block: The panel shows abbreviated Python code rather than standard pseudocode; key functions are truncated and the extensible subclassing mechanism is not illustrated.

(Remarks on code availability)

The drevalpy code is well-organized. It supports the paper's findings effectively. The structure is clear, making it functional for its purpose. The repository does what it aims to do: provide a working benchmark for drug response prediction that others can use.

Version 1:

Reviewer comments:

Reviewer #1

(Remarks to the Author)

The authors fully addressed my comments.

If the authors are interested in providing more insight onto the problem of poor models generalization of the previously unseen drugs in discussion I can recommend the following papers:

Herbert, William G., et al. "Monotherapy cancer drug-blind response prediction is limited to intraclass generalization." bioRxiv (2025): 2025-06.

Narykov, Oleksandr, et al. "Data imbalance in drug response prediction: multi-objective optimization approach in deep learning setting." Briefings in Bioinformatics 26.2 (2025).

Guo, Qianrong, Saiveth Hernandez-Hernandez, and Pedro J. Ballester. "Scaffold splits overestimate virtual screening

performance." International Conference on Artificial Neural Networks. Cham: Springer Nature Switzerland, 2024.

(Remarks on code availability)

All code availability comments were fully addressed

Reviewer #2

(Remarks to the Author)

The authors have addressed all of the concerns raised in my initial review. The revisions improve the clarity and rigor of the manuscript. I have no remaining comments.

(Remarks on code availability)

The code appears well-organized and actively maintained, with a clear directory structure.

Reviewer #3

(Remarks to the Author)

The authors have made extensive and reasonable revisions, which look satisfactory to me.

(Remarks on code availability)

N.A.
